# Sledgehammer to Scalpel: Broad Challenges to the Heart and Other Tissues Yield Specific Cellular Responses via Transcriptional Regulation of the ER-Stress Master Regulator ATF6α

**DOI:** 10.3390/ijms21031134

**Published:** 2020-02-08

**Authors:** Winston T. Stauffer, Adrian Arrieta, Erik A. Blackwood, Christopher C. Glembotski

**Affiliations:** Department of Biology, San Diego State University Heart Institute, San Diego State University, San Diego, CA 92182, USA; wstauffe@gmail.com (W.T.S.); aarrieta1335@gmail.com (A.A.); eblackwo@alumni.nd.edu (E.A.B.)

**Keywords:** ATF6α, ATF6β, ER stress, transcriptional regulation, proteostasis, endoplasmic reticulum, UPR, OASIS, basic leucine-zipper, cardiac

## Abstract

There are more than 2000 transcription factors in eukaryotes, many of which are subject to complex mechanisms fine-tuning their activity and their transcriptional programs to meet the vast array of conditions under which cells must adapt to thrive and survive. For example, conditions that impair protein folding in the endoplasmic reticulum (ER), sometimes called ER stress, elicit the relocation of the ER-transmembrane protein, activating transcription factor 6α (ATF6α), to the Golgi, where it is proteolytically cleaved. This generates a fragment of ATF6α that translocates to the nucleus, where it regulates numerous genes that restore ER protein-folding capacity but is degraded soon after. Thus, upon ER stress, ATF6α is converted from a stable, transmembrane protein, to a rapidly degraded, nuclear protein that is a potent transcription factor. This review focuses on the molecular mechanisms governing ATF6α location, activity, and stability, as well as the transcriptional programs ATF6α regulates, whether canonical genes that restore ER protein-folding or unexpected, non-canonical genes affecting cellular functions beyond the ER. Moreover, we will review fascinating roles for an ATF6α isoform, ATF6β, which has a similar mode of activation but, unlike ATF6α, is a long-lived, weak transcription factor that may moderate the genetic effects of ATF6α.

## 1. Introduction

In eukaryotes, transcription is a highly complex and regulated process involving multiple levels of control. For example, epigenetic regulation of transcription through histone modifications can alter chromatin structure in ways that determine transcriptional magnitude [1]. The positioning of promoter and enhancer sequences in genes governs the recruitment of the polymerases, transcription factors, and other transcriptional machinery [2]. Moreover, transcription initiation, transcript elongation, splicing, and termination are all subject to regulatory checkpoints that serve as determinants of transcriptional programs [3]. The processes that determine what genes are regulated by a given transcription factor are both intricate and varied. At the least, these complex regulatory processes provide a mechanism for the fine-tuning of transcriptional programs involved in numerous responses including development, differentiation, immune responses and responses to stress. Moreover, transcription factors that are broadly expressed can be activated by cell-specific stimuli and they can regulate cell-specific transcriptional programs. Thus, there is a vast and, for the most part, uncharacterized array of genetic responses for even well studied transcription factors with known, or canonical roles. It has been estimated that approximately 8% of the human genome encodes approximately 2600 transcription factors [4], many of which fall into families that are sometimes based on location or structure, such as zinc finger, homeodomain, nuclear hormone receptors, basic helix-loop-helix and basic leucine zipper (bZIP) [5]. One of the most intriguing features of many regulated transcription factors is their mechanism of activation [6]. For example, certain stress-regulated transcription factors are responsive to oxygen depletion (e.g., hypoxia-inducible factor 1α (HIF1α) [7], oxidative stress (e.g., nuclear factor erythroid 2-like factor 2 (NRF2)) [8] and growth factors (e.g., serum response factor (SRF)) [9]. This review focuses on activating transcription factor 6α (ATF6α), a transcription factor that was originally found to be activated by ER proteotoxic stress [10], i.e., protein-misfolding in the ER, but more recently has been found to be activated by a wider array of stresses [11].

## 2. ATF6α Overview

The ER unfolded protein response (UPR) responds to stresses that perturb ER protein folding capacity, i.e., ER stresses that result in the accumulation of misfolded proteins in the ER lumen. ATF6α is a master regulator of one of the three main branches of the ER UPR, each of which is initiated by the ER-transmembrane proteins, protein kinase R-like ER kinase (PERK), inositol-requiring protein-1 (IRE1), and ATF6α (Figure 1A) [12]. In response to misfolded proteins, PERK dimerization results in the activation of its cytosolic kinase function, leading to autophosphorylation and the phosphorylation of numerous proteins outside the ER, including eukaryotic initiation factor 2α (eIF2α) (Figure 1B). eIF2α phosphorylation by PERK confers a global arrest of translation, thus decreasing the protein-folding load on the ER machinery [13]. Upon ER stress, IRE1 also dimerizes and becomes autophosphorylated on its cytosolic domain, which activates it as an RNA splicing enzyme that converts x-box-binding protein 1 (XBP-1) mRNA to a so-called spliced form that encodes an active transcription factor, XBP-1s [14] (Figure 1C). In contrast to PERK and IRE1, when ATF6α senses misfolded proteins in the ER, it translocates to the Golgi (Figure 1D), where it is proteolytically clipped. The N-terminal fragment liberated as a result of this proteolysis [15] is a basic leucine zipper (bZIP) transcription factor related to others in the activating transcription factor/cAMP response element binding (ATF/CREB) family [16,17]. ATF6α was the first of its subgroup of the ATF/CREB family to be identified, and since then a related isoform, ATF6β, which has significant homology to ATF6α, has been assigned to the ATF/CREB family. ATF6β can bind to and transcriptionally induce many of the same genes as ATF6α, but ATF6β is a much weaker transcriptional activator, perhaps serving as an endogenous modulator of ATF6α, as described below [18,19].

## 3. ATF6α Activation

In its transcriptionally inactive state, ATF6α is anchored in the ER partly by binding to the well-characterized ER chaperone, glucose-regulated protein 78 kD (also known as BiP but referred to here as GRP78) [20]. Under these conditions ATF6α forms oligomers via intermolecular disulfide bonds between conserved cysteine residues in its luminal domain [21,22] (Figure 2A). During periods of ER stress, brought on by pathophysiological conditions including ischemia [23], GRP78 is actively diverted to bind to the misfolded proteins that accumulate in the ER lumen, thus releasing its hold on ATF6α [24] (Figure 2B). Subsequently, the disulfide bonds in ATF6α become reduced, which decreases ATF6α oligomerization, leading to the translocation of ATF6α to the Golgi (Figure 2C), where it can then be proteolytically cleaved by regulated intramembrane proteolysis (RIP) by the Golgi resident proteases, site 1 protease (S1P) and site 2 protease (S2P) [25] (Figure 2D). This mechanism of activation involving the release of ATF6α by GRP78 may be shared with PERK and IRE-1, thus fostering the activation of all three branches of the ER stress response [26,27]. However, more recently, other paradigms of ATF6α activation have emerged. For example, thrombospondin 4 (Thbs4), a secreted calcium-binding protein, was shown to interact with ATF6α during stress and to promote its shuttling to the Golgi (Figure 2C,D). In this study, it was shown that without Thbs4, ATF6α activation was blunted, as were the other ER stress branches, suggesting a broader role for Thbs4 in the ER UPR [28]. Recently, multiple protein oxidoreductases have been shown to associate with, and reduce ATF6α during ER stress; this reduction, which also causes the dissociation of ATF6α oligomers, is thought to facilitate the movement of ATF6α out of the ER to the Golgi [21,22]. In particular, the ER-resident protein disulfide isomerase (PDI), PDIA5, was shown to be necessary for the proper reduction of ATF6α before packaging into Golgi-bound vesicles in cancer cells [29]. Also, the PDI, ER protein 18 (ERp18), was found to associate with, and reduce ATF6α during ER stress and, in its absence, ATF6α was improperly processed by the S1P/S2P proteases, which were unable to release its soluble nuclear form [30]. Small molecule compounds have been discovered which take advantage of this reduction/activation dynamic for targeted, specific inhibition or activation of ATF6α. A class of pyrazole amides, dubbed Ceapins, have been discovered [31], which inactivate ATF6α by inhibiting its transport from the ER to the Golgi (Figure 2B). Furthermore, ATF6α that has been retained in the ER by Ceapins is still able to be processed by the Golgi proteases, if those proteases are experimentally relocated from the Golgi to the ER. Indeed, the fact that Ceapins have no effect on S1P/S2P forms the likely basis for why Ceapins are so specific for ATF6α [32]. Other compounds, such as AEBSF [33] or PF429242 [34], also inhibit ATF6α, but do so by inactivating serine proteases, including S1P; thus, a caveat to these compounds is that they impact other proteins that are activated by RIP, such as the transcription factor, sterol regulatory element binding protein (SREBP), as well as some other members of the ATF6 subgroup, including ATF6β (see below) [31,32]. Another chemical recently discovered in a high-throughput screen, is a selective ATF6α activator [35]. This chemical, called compound 147, promotes the formation of ATF6α monomers and thus enhances the transit of ATF6α from the ER to the Golgi and subsequent proteolytic cleavage of ATF6α to its active form [36] (Figure 2B). Compound 147 is specific for ATF6α and does not activate other members of the ATF6 family [35]. Likewise, the naturally occurring sphingolipids dihydrosphingosine (DHS) and dihydroceramide (DHC) have also been shown to specifically activate ATF6α [37] (Figure 2B), though the resulting upregulation of ATF6 target transcripts can vary significantly depending on the activation stimulus (see below). These reagents are very useful for studies requiring selective pharmacologic manipulation of ATF6α activity and, as will be discussed below, may also be of future utility in disease treatment, where ATF6α is a therapeutic target.

## 4. ATF6α Transcriptional Activity and Degradation

Once cleaved, the N-terminal fragment of ATF6α is liberated from the ER and, via a nuclear localization sequence, it translocates to the nucleus where, along with a variety of potential binding partners including itself [38], it can selectively activate transcription (Figure 2E). The transactivation domain (TAD) of ATF6α was found to contain an eight amino acid sequence with homology to the Herpes simplex viral transcription factor, VP16 [39] (Figure 3A,B). This sequence, referred to as VN8 in that same study, has been shown to confer both high transcriptional potency and a high rate of degradation to ATF6α. Amongst the ATF6α subfamily, only ATF6α possesses a VN8 sequence in its transactivation domain. Indeed, while other unrelated transcription factors are also known to be quickly degraded when active [40], ATF6α is the only such mammalian transcription factor known to have a VN8 sequence [39]. Mutating specific residues in the VN8 region of ATF6α is sufficient to significantly reduce its transcriptional potency and significantly increase its stability [18]. It is interesting to note that while ATF6β is relatively similar to ATF6α in most areas it does not have a VN8 domain; consistent with this, ATF6β is a comparatively poor transcription factor and it is degraded very slowly [41] (Figure 3C,D). Moreover, mutation studies demonstrated that transferring the VN8 domain of ATF6α onto ATF6β transformed ATF6β from a long-lived, weak transcription factor into a short-lived, strong transcription factor, resembling the characteristics of ATF6α [18]. The exact mechanism by which activated ATF6α is rapidly degraded remains unknown. While ATF6α has been reported to be ubiquitylated [42,43] and while it is known to be degraded by proteasomes [39,44], less is known about the timing, location, and importance of these events. In so far as the VN8 sequence of VP16 is shared with ATF6α, it is interesting to note that the ubiquitin-proteasome system has been shown to be required for the activity of VP16 [45]. In that study, a specific subunit of the 19S proteasome, 26S proteasome regulatory subunit 8 (SUG-1), was required to ubiquitylate and degrade VP16. Interestingly, this subunit was also required for the transcriptional activity of VP16, suggesting that, at least for VP16, ubiquitylation is a requirement for its transcriptional activity, and is not a consequence of its transcriptional activation [45]. ATF6α has also been shown to be post-translationally modified in other ways, such as glycosylation [46] and SUMOylation [47], and it is certainly true that many post-translational modifications, including ubiquitylation, have been reported to be essential regulators of other transcription factors [48]. Meanwhile, there are varying reports concerning whether nuclear proteasomes are functional and whether ubiquitylated nuclear proteins, potentially including ATF6α, are degraded in the nucleus, or whether they are first transported back to the cytosol [49,50,51].

## 5. ATF6α Dimerization and Nuclear Binding Partners

ATF6α, like other bZIP transcription factors, forms homodimers, as well as heterodimers with other transcription factors, not all of which are members of the ATF/CREB family; presumably, it is through selective dimerization that the transcriptional programs regulated by ATF6 can be fine-tuned [52] (Figure 2E). ATF6α was first discovered as a dimerization partner of serum response factor (SRF), a MADS-box transcription factor that upregulates a variety of genes through serum response elements (SRE) [53]. SRF binding to ATF6α was shown to enhance the activation of SRF-responsive transcript expression [54]. ATF6α binding to its target DNA sequences is also dependent on interactions with other proteins, such as NF-Y, which also binds these gene targets [55]. Other binding partners that synergistically enhance the activity of ATF6α include YY-1 [56], fellow ER stress response effector XBP-1s [57], PGC-1α [58], ERRγ [59], and CREBH [60] (see below). Additionally, binding partners have the potential to negatively regulate the transcriptional activity of ATF6α, as is the case with ATF6β [41] (see below). The mechanisms that determine ATF6α binding partners remain unknown. It seems unlikely that ATF6α homo- or heterodimerization could begin in the ER because, as before, only monomeric ATF6α is packaged into COPII vesicles for exit from the ER [22]. Dimerization could take place in the Golgi, though when cleavage of ATF6α is experimentally forced to take place in the ER, using mutant forms of S1P/S2P proteases engineered with C-terminal KDEL ER retention sequences, ATF6α is processed normally and is functional [61]. Lastly ATF6α may find its binding partners after translocation to the nucleus, perhaps in the process of binding DNA sequences. Indeed, in the case of bZIP-bZIP dimers, the dimer structure of the combined transcription factors is required for proper binding to the major groove of the DNA [62].

## 6. ATF6α Promoter Elements

The specific promoter sequences to which ATF6α binds were first reported by Kazutori Mori’s group and were named ER stress response elements (ERSEs) [63]. In that study, it was shown that the consensus sequence of ERSEs is CCAAT-N_9_-CCACG, which includes the binding sites for both NF-Y (CCAAT) and the ATF6α dimer itself (CCACG) with a nine-nucleotide spacer in between [63]. Subsequent studies identified variations of this sequence, as well as an entirely new ERSE, ERSE-II, the sequence of which, AATTG-N-CCACG, was also found to be important for the ATF6α transcriptional program [64].

## 7. ATF6α Transcriptional Programs

In the heart, ATF6α induces at least 400 gene transcripts in vivo; many of these genes have ERSE or ERSE-like elements [65]. RNAseq performed on mouse hearts expressing a tamoxifen-activatable form of ATF6α revealed a wide range of upregulated transcripts involved in many critical processes [66]. As one would expect, many of these gene products are directly involved in the adaptive ER stress response and act to enhance ER protein folding capacity under adverse conditions. These gene products include those that encode ER luminal chaperones like GRP78 and GRP94 [63,67] or PDIs like PDIA6 [68] which, in addition to roles in the activation of the ER stress sensors, bind nascent proteins in the ER and promote their correct folding into secondary and tertiary structures by facilitating disulfide bond formation [68]. Another subset of ATF6α-regulated genes is involved in ER-associated degradation (ERAD) pathways. Terminally misfolded proteins cannot be repaired and must therefore be targeted for degradation before they build up and become potentially toxic. Because there are no proteasomes in the ER lumen, these proteins must be transported to the cytosol for degradation [69,70]. Intimately involved in this process is the ATF6α-inducible gene product, HMG-CoA reductase degradation protein 1 (Hrd1), an ER transmembrane E3 ubiquitin ligase that both transports misfolded proteins out of the ER and ubiquitylates them, after which they are degraded by proteasomes on the cytosolic face of the ER [71,72]. Interestingly, Hrd1 was the only transmembrane E3 ubiquitin ligase induced by ATF6α in the heart [65,73]. Other ERAD components that are induced by ATF6 include degradation in ER protein 3 (Derlin3) [74] and VCP-interacting membrane protein (VIMP) [75]. These proteins preserve ER proteostasis and enhance cell survival [76,77] and thus, ATF6α is commonly associated with the adaptive ER stress response [78,79]. However, persistent ER stress and ATF6α activation can activate maladaptive pathways, which guide the cell towards apoptosis [80,81]. Amongst these maladaptive ATF6α transcriptional targets is CHOP [82,83,84], which induces apoptotic signaling, showing that ATF6α is not always adaptive in a protective sense. Other candidates revealed in RNAseq datasets as ATF6α-regulated genes in the heart, and confirmed in recent discoveries, are non-canonical ATF6α transcriptional targets that are not directly involved in ER protein homeostasis. For example, the well-known antioxidant enzyme catalase, which is found in peroxisomes and not in the ER, is induced by ATF6α in the heart [75]. Therefore, by inducing catalase, ATF6α can reduce the levels of reactive oxygen species (ROS), which has functional implications far beyond ER proteostasis. Another example of a non-canonical ATF6-regulated gene product is the small GTPase Ras homolog enriched in brain (Rheb), which, when induced by ATF6α in the heart, activates mammalian target of rapamycin complex 1 (mTORC1) and thus drives protein synthesis and hypertrophic cardiac myocyte growth [66]. Again, this indirectly affects ER protein folding but further shows how ATF6α can be a master regulator of global cellular responses.

## 8. Stimulus-Specific ATF6α Transcriptional Programs

Recently, it has been found that a wide range of cellular stresses can activate ATF6α; intriguingly, the transcriptional programs regulated by ATF6α appear to be stress specific [66]. For example, the canonical ATF6α-target gene product, GRP78, which is directly involved in protein folding in the ER, is induced by conditions that acutely increase ER protein misfolding (Figure 4A); however, the more recently identified ATF6α-regulated gene product, catalase, a peroxisomal anti-oxidant enzyme, is induced mainly by ATF6α when it is activated by oxidative stress, but not when ATF6α is activated by ER protein misfolding (Figure 4B). Finally, Rheb, which is not induced by ATF6α when it is activated by oxidative stress, is induced by ATF6α when it is activated by growth signals; under these conditions, which were studied in the heart, ATF6α was required for cardiac myocyte growth in response to phenylephrine treatment (Figure 4C). Another recent study by Tam et. al. revealed that ATF6 is selectively activated by the sphingolipids dihydrosphingosine (DHS) and dihydroceramide (DHC) [37]. Like treatment with compound 147, activation is limited to ATF6α and there is no general induction of ER stress (Figure 2B). Uniquely, however, this mode of activation causes ATF6α to induce a previously unknown ATF6α-regulated transcriptional program involved in enhancing lipid production for membrane expansion [37]. Given that ATF6α activation in all these scenarios still appears to involve the canonical ER-Golgi-nucleus pathway, it is unclear how each stimulus directs ATF6α to a specific subset of genes. It is likely that there are critical roles being played by thus far unknown post-translational modifications, binding partners, and epigenetic regulators which are induced by each stimulus and guide ATF6α to the appropriate gene set.

## 9. ATF6α in Disease

Transcriptional regulation of a transcriptional program as extensive as that mediated by ATF6α naturally has profound consequences in a variety of disease states. For example, protein aggregation is a common feature of cardiomyopathy and, in numerous studies of ATF6α in the heart, it has been consistently found to play protective roles in multiple forms of cardiac injury [23,28,66,75,85,86,87]. Martindale et. al. overexpressed a tamoxifen activatable form of ATF6 in mouse hearts and then performed ex vivo global ischemia reperfusion on the isolated mouse hearts. Hearts with activated ATF6α had significantly smaller infarcts and improved cardiac function compared to control [85]. In a later study, Lynch et. al. investigated Thbs4 in hearts from mutant mice that constitutively develop a buildup of cardiac misfolded protein aggregates. They found that Thbs4 was protective against such aggregates because of its role in the activation of ATF6α [28]. More recently, Jin et. al. performed ischemia/reperfusion surgeries in mice, where damage to the heart is due mainly to the generation of ROS upon reperfusion. Endogenous ATF6α was found to be activated by this injury and was shown to be protective because it induces the antioxidant, catalase [75]. Blackwood et. al. discovered that ATF6α was necessary for compensatory cardiac growth in response to pressure overload induced by transaortic constriction (TAC) surgeries, which mimic chronic high blood pressure, due to its induction of the mTORC1 activator Rheb [66]. In a subsequent publication, exploring the use of compound 147 as a drug, Blackwood et al. found 147 to be globally protective against ischemia/reperfusion models in the heart, kidney, and brain [87]. ATF6α and the ER stress response are generally found to be highly activated in professional secretory cells, due to the increased protein flux through the ER. Sharma and colleagues accordingly found ATF6α contributes to beta cell proliferation in the pancreas, with implications for diabetes [88]. The ER stress response is also highly active in cancer cells, which, in addition to rapid proliferation, must also survive in the hypoxic environment inside tumors. Indeed, ATF6α confers chemoresistance in leukemia cells [29]. Lastly, naturally occurring ATF6α mutations have been found in humans that can alter its activation in response to ER stress. Certain single nucleotide polymorphisms that increase the amount of ATF6α signaling have been associated with increases in blood cholesterol in patients at risk of cardiovascular disease, potentially because ATF6α may interact with the SREBP2 lipid biosynthesis pathway [86]. Other mutations that lead to the truncation and degradation of ATF6α transcript have been shown to cause the autosomal recessive eye disorder achromatopsia, which is characterized by non-functional cone cells, resulting in color-blindness, photophobia, and other maladies [89].

## 10. ATF6α Relatives

While the importance of the varied and complex outcomes of ATF6α signaling is apparent in multiple tissues and disease states, it is important to note that ATF6α is related to other transcription factors, many of which are often included in the large ATF/CREB family [16,17]. Like ATF6α, all members of the ATF/CREB family are bZIP transcription factors consisting of a basic DNA binding region next to an alpha-helical coiled coil region for dimerization. While it is this domain that is the basis of their grouping as a family, the wider structure, distribution, mode of activation, and functions of these family members can vary greatly [52,62]. Amongst these ATF/CREB family members, only a subset are type II ER-transmembrane transcription factors that, like ATF6α, are activated by proteolytic cleavage by Golgi-resident S1P/S2P proteases. In the literature, this subfamily is alternatively named for ATF6 [90], CREB3 (Luman) [91], or, as in this review, for old astrocyte specifically induced substance (OASIS) [92]. The OASIS subfamily includes Luman, primarily expressed in ganglionic neurons, monocytes, and dendritic cells [93]; OASIS, expressed in osteoblasts and astrocytes and involved in bone development [94,95]; box B-binding factor 2 human homolog on chromosome 7 (BBF2H7), expressed in chondrocytes and critical for cartilage development [95,96]; cAMP response element binding H (CREBH), expressed in the liver and known to heterodimerize with ATF6α [60]; CREB4, expressed in the prostate and intestines [97]; and ATF6β discussed below. In all cases the liberated cytosolic N-terminal fragment is a bZIP transcription factor with a high degree of structural homology to the activated form of ATF6α, including an N-terminally located TAD and a C-terminal bZIP DNA binding/dimerization domain [16,90,92]. However, none have a TAD sequence that is homologous to that of ATF6α [39] (Figure 5). Additionally, the luminal regions of the rest of the OASIS subfamily (except for ATF6β) do not resemble ATF6α. In particular, some lack GRP78 binding domains and do not have apparent Golgi localization sequences [92]. Thus, the specific mechanism of activation for some family members is obscure. Many OASIS members have tissue specificity in terms of their expression, while ATF6α is broadly expressed in all tissues [92,95]. OASIS members also bind to differing promoter sequences. Some are known to bind to ERSE, unfolded protein response elements (UPRE), or otherwise ATF6-like sequences, while others show preference to CRE or CRE-like promoter elements [90,92]. It is unclear to what degree OASIS members are involved in ER stress. Some are activated by ER stress and some are not [90,91,92]. Like the rest of the ATF/CREB family, they form homodimers in order to bind DNA and activate transcription and have the potential to heterodimerize with similar transcription factors, including ATF6α [90,92,98].

## 11. ATF6β

In terms of amino acid sequence, ATF6β is the most closely related OASIS subfamily member to ATF6α and the two together are sometimes considered to form their own subfamily [16,19]. Like ATF6α, ATF6β is an ER transmembrane protein with an N-terminal bZIP domain that is liberated by Golgi proteases upon ER stress [20]. ATF6β has significant sequence homology to ATF6α, not only in its bZIP and transmembrane domains, which it shares with the OASIS subfamily members (Figure 5), but also in other areas of its cytosolic and luminal domains [19,20]. Again, however, one area of noted difference is in its TAD, which lacks the critical eight amino acid VN8 sequence found in ATF6α [18]. When activated, ATF6β is a relatively weak transcriptional activator with a long half-life. Because ATF6β can both bind the same ERSE regions and form heterodimers with ATF6α, it can thus slow the activity of α and function as a transcriptional repressor [18,41,99]. However, it has more recently been shown to upregulate at least some of the same transcripts as ATF6α. Given its greater stability it thus may be able to compensate in model systems in which ATF6α has been deleted, a hypothesis that agrees with the fact that while either ATF6 α or β knockout mice survive to adulthood, double-knockouts are embryonic lethal [100].

## 12. Conclusions

Though ATF6α is just one transcription factor in the ATF/CREB family, and among more than 2000 transcription factors [4,101] encoded in the mammalian genome, it provides an illuminating case study highlighting the complex regulation that can occur with such a tightly controlled master switch for multiple signaling pathways. Because ATF6α is so essential to cell survival and function during numerous environmental stresses, and because it is so highly conserved and ubiquitously expressed amongst eukaryotes, it becomes clear why it is subject to such finely tuned control networks. Understanding the intricacies of each step in ATF6α activation can provide insights, and perhaps lead to the identification of potentially druggable regulatory circuits for treating a variety of human diseases. Care must be taken to avoid painting ATF6α and other similarly important transcriptional regulators with too broad a brush, pigeonholing them into “known” or “canonical” pathways. New and complex regulatory circuits are continually being uncovered and explored, likely revealing undiscovered roles these signal transducers play. As is so often the case, when discussing transcriptional regulation, the devil is in the details.

## Figures and Tables

**Figure 1 ijms-21-01134-f001:**
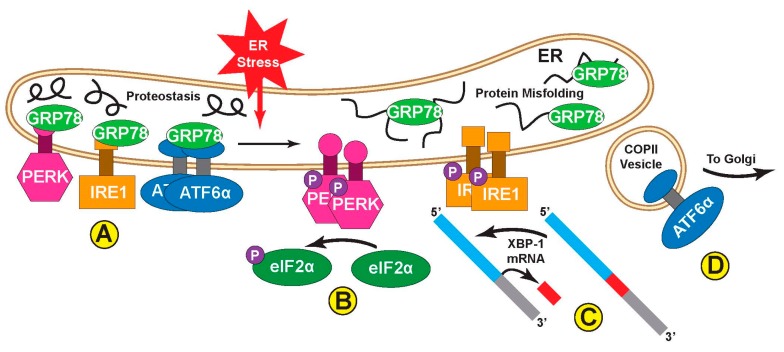
Overview of the activation of the three branches of the ER UPR. (**A**) PERK, IRE1, and ATF6α are all ER transmembrane proteins with cytosolic and luminal domains that act as sensors to detect the build-up of misfolded proteins characteristic of ER stress. When the capacity of the ER to translate and fold nascent proteins is sufficient to meet demand, protein misfolding is minimal and the ER is in proteostasis. During such periods, canonical ER chaperone GRP78 (also known as BiP) binds the luminal domains of all three UPR sensors and helps hold them in an inactive state. PERK and IRE1 are held in monomeric form, while inactive ATF6α is an oligomer. Upon induction of ER stress, by challenges such as ischemia, oxidative stress, or increases in ER protein flux, nascent ER proteins begin to misfold and can build up in the ER lumen. GRP78 then actively unbinds the ER sensors to bind the misfolded proteins and promote their proper folding. Simultaneously, the three sensor branches become active; (**B**) PERK, which has cytoplasmic kinase domains, dimerizes with itself and becomes autophosphorylated. It then phosphorylates its substrate, eIF2α, in the cytoplasm which acts to slow translation of all non-UPR-related proteins; (**C**) IRE1, which has cytoplasmic kinase and endonuclease domains, also dimerizes and is autophosphorylated. This activates its endonuclease function to splice out a section of XBP-1 mRNA to create a new, spliced transcript, XBP-1s, which codes for a transcription factor that upregulates some UPR-related transcripts; (**D**) ATF6α, which is both a sensor of ER stress and a UPR-effector transcription factor, becomes monomeric upon induction of ER stress, whereupon it exits the ER via COPII vesicles and transits to the Golgi, where the active transcription factor is liberated from the transmembrane domain by Golgi proteases.

**Figure 2 ijms-21-01134-f002:**
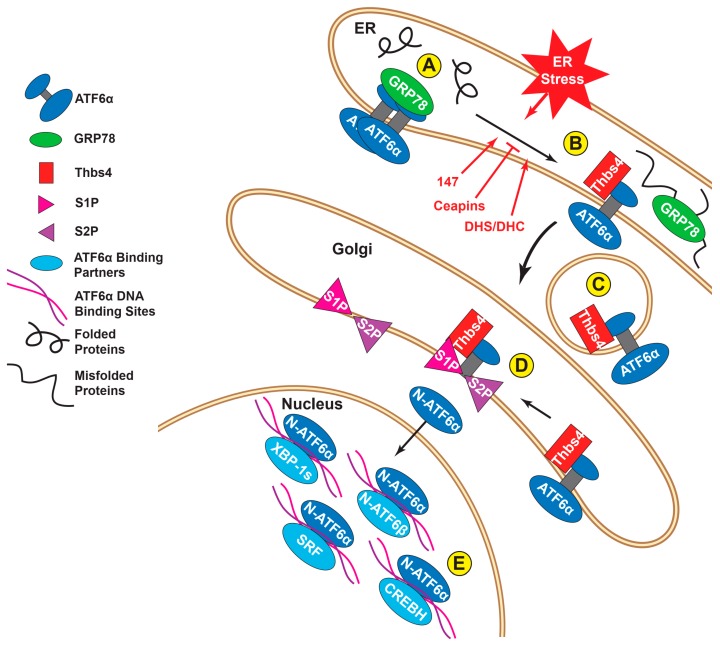
Focus on ATF6α activation. (**A**) In the absence of ER stress, ATF6α exists in an oligomeric state and its luminal domain associates with GRP78; (**B**) During ER stress, GRP78 relocates from ATF6α to misfolded proteins; ATF6α is then reduced to a monomeric state. Chemicals, such as compound 147, or certain sphingolipids, can promote ATF6α relocation out of the ER in the absence of overt ER stress. Other chemicals, called Ceapins, inhibit this process, even during ER stress; (**C**) ATF6α and binding partners, such as Thbs4, are packaged into COPII vesicles and transit to the Golgi; (**D**) Golgi resident proteases, S1P and S2P, cleave the N-terminal cytosolic region of ATF6α from the transmembrane domain. The soluble N-terminal fragment (N-ATF6α) is then free to enter the nucleus where it binds to target genes and influences transcription; (**E**) N-ATF6α binds DNA as a homodimer but is also known to form heterodimers with a number of other nuclear proteins, including N-ATF6β, XBP-1s, SRF, and CREBH. Binding different partners can alter which transcripts N-ATF6α induces.

**Figure 3 ijms-21-01134-f003:**
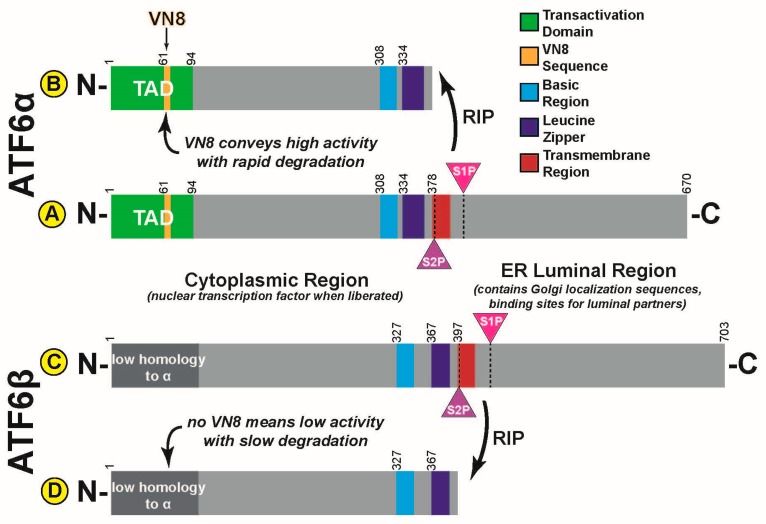
Structural comparison between ATF6 α and β. (**A**) Full length ATF6α has an N-terminal cytosolic region with a transcriptional activation domain (TAD), a central basic leucine zipper domain (bZIP), and a transmembrane domain followed by a C-terminal luminal region; (**B**) The cytosolic region is liberated from the ER membrane by S1P and S2P Golgi proteases in a process called regulated intramembrane proteolysis (RIP). The TAD of the N-terminal fragment contains an eight-amino acid sequence, from amino acids 61 to 68, called the VN8, which confers high transcriptional activity and rapid degradation; (**C**) Full length ATF6β has a similar structure to ATF6α and its cytosolic region is also liberated from the ER membrane by S1P and S2P mediated RIP; (**D**) The N-terminal region of the soluble portion of ATF6β has low homology to ATF6α and, in particular, lacks the VN8 domain. N-ATF6β is thus a weak transcriptional activator with a long half-life.

**Figure 4 ijms-21-01134-f004:**
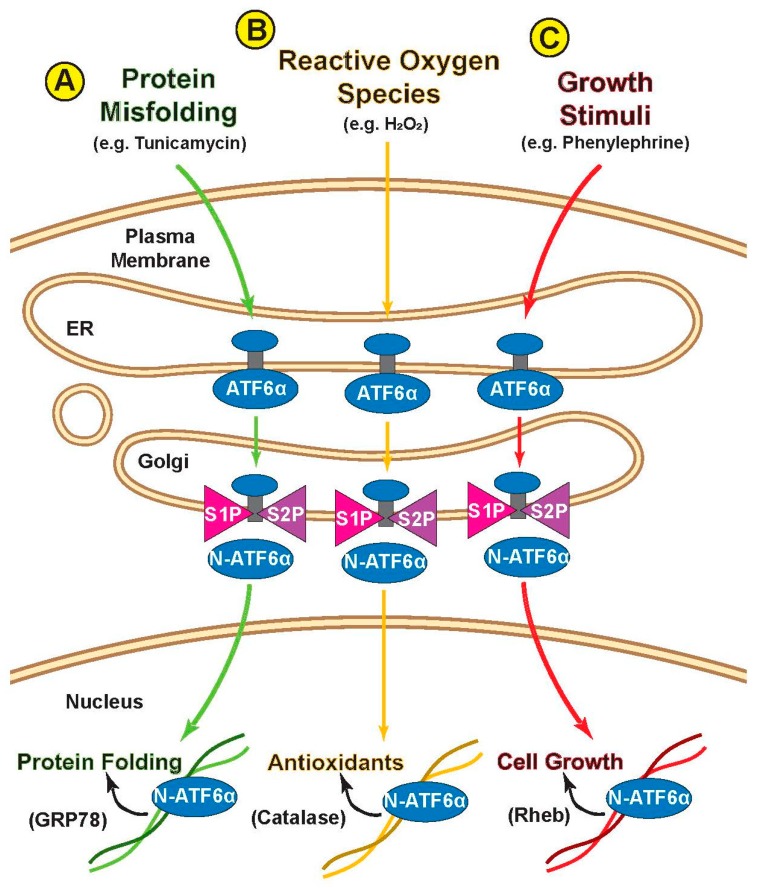
Stimulus-specific transcriptional programs for ATF6α. Different stimuli cause ATF6α to induce different transcriptional targets, depending on the stimulus, despite activating ATF6α in apparently similar ways. (**A**) N-ATF6α induces canonical ER chaperone GRP78 in response to protein misfolding; (**B**) N-ATF6α upregulates antioxidant catalase, but only in response to oxidative stress; (**C**) N-ATF6α upregulates cell growth inducer Rheb, but only in response to growth signals.

**Figure 5 ijms-21-01134-f005:**
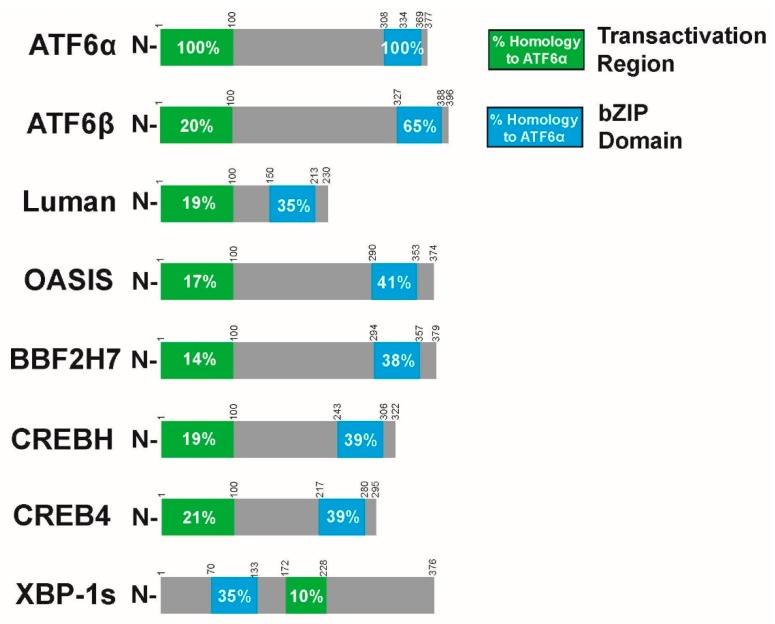
Sequence homologies to soluble N-terminal ATF6α amongst OASIS family members. While all OASIS members have similar structure in their N-terminal fragments, they have low sequence homology to ATF6α in their TADs. There is greater homology, however, in their bZIP domains, especially for the member most closely related to ATF6α, ATF6β. Fellow ER stress effector XBP-1s, which is a bZIP transcription factor but is not in the OASIS subfamily, is included here for comparison.

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
