# Peer review of "Sledgehammer to Scalpel: Broad Challenges to the Heart and Other Tissues Yield Specific Cellular Responses via Transcriptional Regulation of the ER-Stress Master Regulator ATF6α"

_ijms, 2020, doi:10.3390/ijms21031134_

Round 1
Reviewer 1 Report
Stauffer et al review the activities of the ER stress-responsive transcription factor ATF6, within describing the ER stress response, the regulation of ATF6, the transcriptional targets of ATF6, and the roles ATF6 orchestrates in physiology (among others). Throughout, there is a recurring theme that transcription factors perceived as general or canonical such as ATF6 can play precise roles in maintaining homeostasis during normal physiological states and in disease. The authors carefully explore this idea, demonstrating a number of cases in which unique stressors and cell states result in specific ATF6-dependent transcriptional adaptations. Importantly, this manuscript also addresses the druggability or pharmacological ligandability of ATF6 (and the ERSR), an emerging theme from the literature relevant to human health. Overall, this work is comprehensive and informative and is not omitting any key concepts from this field. The organization of the manuscript is logical, and the figures accurately and succinctly depict the discussed subject matter. This manuscript will make a fine contribution to IJMS, and I fully support its publication without major revision. Below, I suggest minor changes to the text before its publication.
Issues:
In general, there is a reliance upon unclear terms, like “gene programs” for referring to “transcriptional programs.” Overall, the careful use of precise nomenclature throughout would benefit the clarity and formality of the work (e.g., line 221 would be better served as 400 transcripts rather than 400 genes, etc.). For a review specifically addressing transcription, the authors must be careful with this language. GRP78 is frequently called BiP in the literature and this nomenclature is ignored in this manuscript. While I understand the desire to do so, this is standard for the field and addressing this in some way (“also known as BiP”) at least once in text and once in figure will help with any uncertainty to those unfamiliar with this field. Line 6: “transcriptional rate” should be expressed as a clearer term such as “transcriptional magnitude” Figure 3 lower (ATF6beta): “low homology to alpha” label clashes with lower TAD designation and is unclear. This could be addressed by placing “low homology to alpha” above or below this sequence with arrow or bracket (e.g., VN8 in upper ATF6alpha portion of figure). Line 284 should likely read, “…a tamoxifen activatable form of ATF6…” Line 201 should like read, “PGC1alpha” Line 414 should likely read, “PGC1alpha” Line 370: There is ambiguity in the term “targets.” What kind of targets? Transcriptional outputs of ATF6? Protein drug targets? Fig 2 and Fig 4: Bolded “Figure 2” and “Figure 4” should be removed the body of the figure panel, as Figure 1 and 3 do not have this. Figure 4 upper (under (C)) should likely read “growth stimuli” Figure 4 lower “(GRP78)”, “(Rheb)”, and “(Catalase)” should likely read (e.g., GRP78) as above for Tunicamyin and others
Reviewer 2 Report
This is a needed and very comprehensive review of the ATF6 branch of the UPR. I learned a lot and so will anyone else who reads it! The only comment I will make about the content is that there is a section on small molecules that affects ATF6 activation, and in that the natural small sphingolipids described in the work of the Niwa lab are not mentioned. I thought this was an omission, but they turned up later in the review. I suggest to add a line in the first mention of small molecule modifiers of ATF6 including the sphingolipids and the rejoinder that they will be described in detail below. Something like that. But this is not a deal breaker. Great job!
